

# Identification and analysis of immunoreactive proteins of *Shigella flexneri* in human sera and stool specimens

Kirnpal Kaur Banga Singh[1], Mohd Zaki Salleh[2], Naveed Ahmed[1], Chan Yean Yean[1] and Asma Ismail[3]

[1] Department of Medical Microbiology and Parasitology, School of Medical Sciences, Health Campus, Universiti Sains Malaysia, Kubang Kerian, Kelantan, Malaysia
[2] Integrative Pharmacogenomics Institute (iPROMISE), UiTM Puncak Alam Campus, Bandar Puncak Alam, Puncak Alam, Selangor, Malaysia
[3] Institute for Research in Molecular Medicine (INFORMM), Universiti Sains Malaysia, Kubang Kerian, Kelantan, Malaysia

## ABSTRACT

**Background:** The method currently available to diagnose shigellosis is insensitive and has many limitations. Thus, this study was designed to identify specific antigenic protein(s) among the cell surface associated proteins (SAPs) of *Shigella* that would be valuable in the development of an alternative diagnostic assay for shigellosis, particularly one that could be run using a stool sample rather than serum.
**Methods:** The SAPs of clinical isolates of *S. dysenteriae*, *S. boydii*, *Shigella flexneri*, and *S. sonnei* were extracted from an overnight culture grown at 37 °C using acidified-glycine extraction methods. Protein profiles were observed by SDS-PAGE. To determine if antibodies specific to certain *Shigella* SAPs were present in both sera and stool suspensions, Western blot analysis was used to detect the presence of IgA, IgG, and IgM.
**Results:** Immunoblot analysis revealed that sera from patients infected with *S. flexneri* recognized 31 proteins. These SAP antigens are recognized by the host humoral response during *Shigella* infection. Specific antibodies against these antigens were also observed in intestinal secretions of shigellosis patients. Of these 31 *S. flexneri* proteins, the 35 kDa protein specifically reacted against IgA present in patients' stool suspensions. Further study illustrated the immunoreactivity of this protein in *S. dysenteriae*, *S. boydii*, and *S. sonnei*. This is the first report that demonstrates the presence of immunoreactive *Shigella* SAPs in stool suspensions. The SAPSs could be very useful in developing a simple and rapid serodiagnostic assay for shigellosis directly from stool specimens.

Corresponding author
Kirnpal Kaur Banga Singh,
kiren@usm.my

## INTRODUCTION

Shigellosis, or bacillary dysentery, remains an unsolved problem, especially in developing countries where it continues to affect large numbers of people despite modern medical treatment (*Zhang et al., 2020*). An estimated 164.7 million cases of shigellosis occur throughout the world each year, with 1.1 million deaths (*Kotloff et al., 2018*;

*Qasim et al., 2022*). About 90% of global shigellosis is attributed to *S. sonnei* (*Clarkson et al., 2020*; *Venkatesan et al., 2021*) and *S. flexneri* (*Brengi et al., 2019*). *S. dysenteriae* causes the most severe infection that leads to high fatality (upto 20%) (*Jalal et al., 2022*).

The molecular origins of *shigella* pathogenesis have been extensively discussed previously on a large scale. After being ingested orally, *shigella* passes through the hostile intestinal microbiota and enters into the terminal ileum, colon, and rectum, where it penetrates the mucous layer *via* a variety of bacterial effector proteins, such as IpaA-D and VirG/IcsA (*Qasim et al., 2022*; *Zhang et al., 2021*). A virulence plasmid shared by all *shigella* species encodes these effectors as well as the needle-like type III secretion mechanism that delivers them to the host cell cytoplasm (*Bhaumik et al., 2023*).

The method currently available to diagnose shigellosis involves bacterial culture of stool specimens or rectal swabs. However, the culture method has a low isolation rate and is time consuming and labor intensive (*Ali et al., 2022*; *Maczuga, Tran & Morona, 2022*). Moreover, this bacterium may not grow in stool cultures due to delayed in sample transportation and processing or due to the prior use of antibiotic treatment (*de Alwis et al., 2021*; *Gupta & Dhaked, 2019*). In case of shigellosis infections in young children (<5 years old), the direct detection using a stool specimen is preferred as it is non-invasive, relatively easy to collect, and provides direct evidence of the infection (*Halimeh et al., 2021*; *Kapoor et al., 2022*).

We believe that identification of specific antigenic proteins of *Shigella* is a necessary step in developing the new diagnostic test for shigellosis. Cell surface associated proteins (SAPs) are the prime candidates for recognition by host antibodies because of their exposure on the surface of the cell. In this study, immunoblot analysis was used to confirm the existence of immunogenic and specific SAPs of *Shigella* spp. in sera and in stool suspensions.

## MATERIALS AND METHODS

### Ethical clearance
The study was conducted in accordance with the Declaration of Helsinki and approved by the Human Research Ethics Committee of Universiti Sains Malaysia (protocol code: USMKK/PPSP/JEPeM [248.3(10)]) for studies involving human samples. Written informed consent was obtained from all subjects involved in the study. Written informed consent has been obtained from the patients to publish this article.

### Study design and setting
This cross-sectional study was conducted at the Department of Microbiology and Parasitology, School of Medical Sciences, Universiti Sains Malaysia (USM).

### Bacterial strains
Clinical strains of *S. flexneri* (SF480), *S. dysenteriae* (SD-USM1), *S. sonnei* (SS-USM1), and *S. boydii* (SB-USM1) were obtained from the Diagnostic Laboratory, Department of Medical Microbiology & Parasitology, School of Medical Sciences, USM, Kubang Kerian, Kelantan, Malaysia. The bacterial isolates were shifted to 20% TSB-glycerol and stored at −20 °C. Before starting the wet lab work, working cultures were prepared from the stool

by inoculating a single isolated colony into 10 ml of nutrient broth for experimental purposes. These cultures were then overnight incubated at 37 °C with 200 rpm in a shaking incubator, and then were subcultured on blood agar to ensure the purity.

## Sera and stool specimens

Sera and stool specimens were collected from consented patients admitted to the Hospital USM, Kubang Kerian, Kelantan, Malaysia. Sera and stool specimens were collected from the patients who were infected with *S. flexneri* (culture confirmed) and from patients infected with other enteropathogens common in this region (*Salmonella typhi*, enteropathogenic *Escherichia coli*, *Campylobacter jejuni*, *Vibrio cholerae*, *Giardia lamblia*, and *Entamoeba histolytica*). All sera and stool specimens were collected within 4 to 7 days from onset of symptoms.

The sera and stool specimens (more specifically the antibodies present in them) were used as probes against the SAPs isolated from the bacteria (see next section). Sera samples were used in 1/100 dilutions, whereas a 10% stool suspension was generated to detect presence of specific immunoglobulin(s) against the SAPs. Stool suspension was prepared by thoroughly mixing approximately 100 mg (a pea-sized aliquot) of semi-solid stool or 100 μl of watery stool with 1 ml of TBS buffer. The mixture was then centrifuged at 8,000 $g$ for 2 min. The supernatant was transferred to a sterile tube and then was ready for use in the immunoblot assay. Freshly prepared stool supernatants were used in this study.

## Isolation of SAPs

Crude SAPs were extracted from *S. flexneri* (SF480), S. *dysenteriae* (SD-USM1), *S. sonnei* (SS-USM1), and *S. boydii* (SB-USM1) using glycine-HCL (0.2 M) (pH 2.2). For each of the bacterial species, the colonies were grown in nutrient broth and incubated for 18 h at 37 °C in a shaker incubator. Bacterial cells were harvested (*Harikrishnan, Banga Singh & Ismail, 2017*), washed, and suspended in 30 mM Tris pH 7.4. The wet weight of the bacterial pellet was estimated, and 500 μl of 0.2 M glycine-HCL (pH 2.2) were added to every 0.03 g of the pellet. The suspension was immediately vortexed for 30 s and placed on ice for 30 s. This process was repeated for 10 min. The cell debris was pelleted by centrifugation at 12,000 $g$ for 10 min at 4 °C. The supernatant was carefully transferred to a new tube and immediately neutralized to pH 7.4 with 3 M NaOH. The proteins were then precipitated by adding two volumes of ice-cold ethanol (100%) and kept at −20 °C overnight. Finally, the precipitated proteins were pelleted by centrifugation at 12,000 $g$ for 20 min at 4 °C and then resuspended with 30 mM Tris pH 7.4. An ESL spectrophotometric assay kit (Boehringer Mannheim, Ridgefield, CT, USA) was used to check the protein concentration with bovine serum albumin as the standard.

## Profiling of the SAPs using sodium dodecyl sulfate polyacrylamide gel electrophoresis

Sodium dodecyl sulfate polyacrylamide gel electrophoresis (SDS-PAGE) was done for analyzing proteins based on differences in their molecular size. The partially purified SAPs were solubilized with SDS and β-mercaptoethanol and resolved by SDS-PAGE using 10%
acrylamide under reducing conditions using discontinuous buffer systems (*Kotloff et al., 2018*). 15 µg of the SAPs was loaded in each well of the gel (175 µg of SAPs was loaded in a preparative gel) and run at a constant current of 30 mA at 4 °C for 4 h using a Bio-Rad Protean II™ Slab Cell. The SAPs profiles resolved on SDS-PAGE gels were visualized *via* staining with Coomassie Brilliant Blue (Bio-Rad, Hercules, CA, USA). The locations of the proteins of interest were determined using molecular weight markers (Pharmacia, Peapack, NJ, USA).

## Determining the presence of antigenic and specific protein(s) of *S. flexneri* in SAPs using western blot analysis

To determine the presence of antigenic and specific protein(s) of *S. flexneri*, western blot was performed. The SAPs separated *via* SDS-PAGE were electrophoretically transferred onto a 0.45 µm nitrocellulose membrane using a semi-dry transfer cell (Trans-blot® SD, Bio-Rad, Hercules, CA, USA). At room temperature, 5% skim milk was used to block the membrane for 30 min on a shaker. The blocked nitrocellulose membrane was cut into strips and incubated for 4 h with 1:100 dilutions of 10% freshly prepared stool suspensions or human sera. The antibody-treated nitrocellulose strips were then washed six times for 10 min with 0.05% of PBS-Tween 20. Membrane strips were then incubated for 2 h with alkaline phosphatase-conjugated anti-human IgG (1:1,000), IgM (1:1,000), and IgA (1:500) purchased from Sigma (Burlington, MA, USA). The strips were washed again six times for 10 min with 0.05% PBS-Tween 20 (0.05%). Immunoreactive bands were visualized by incubating the strips for 30 min with alkaline phosphatase color development substrate (Bio-Rad, Hercules, CA, USA). An image analyzer was used to determine the bands' molecular weight by comparing the bands' molecular weight to reference standards. Later research revealed the ideal protein candidate to be a biomarker for the development of diagnostic tests.

# RESULTS

## Determination of the optimum time period for the extraction of *Shigella* surface proteins

The results of the surface protein profile in Fig. 1 shows that the surface protein band at 5 min was almost the same as the surface protein profile at other test times. These results were based on the number of protein bands and the intensity of Coomassie staining produced. This determination was repeated three times and almost the same results were obtained. Therefore, the optimal time chosen for the preparation of surface protein extract in the next study was 5 min.

## The response of *Shigella flexneri* infection sera compared with other infection sera against the surface protein of *Shigella flexneri* strain SF480

Western blot analysis revealed specific IgA, IgM, and IgG responses against the SAP antigens in the sera and stool suspensions. A total of 31 antigenic proteins were recognized

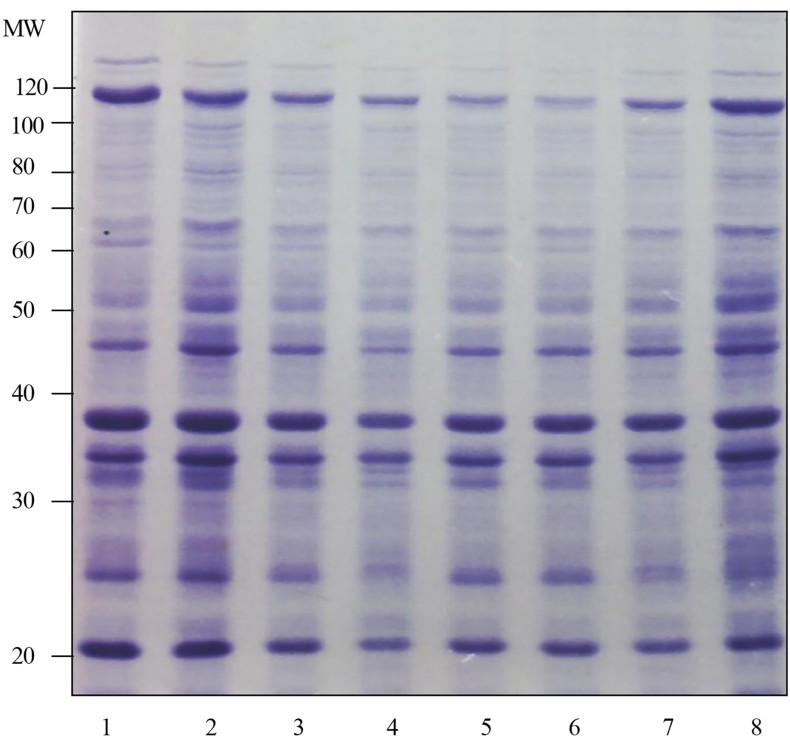

**Figure 1 Determination of the optimum time for the preparation of surface protein extract of *Shigella flexneri* SF480.** Lanes 1 to 8 show the preparation times of *Shigella flexneri* SF480 surface protein extracts. Lane 1: 5 min, lane 2: 10 min, lane 3: 15 min, lane 4: 20 min, lane 5: 25 min, lane 6: 30 min, lane 7: 45 min, lane 8: 60 min (Separation gel = 10%).

when probed with sera from those patients who were infected with *S. flexneri*. Most of these antigenic bands were also recognized by sera from patients infected with the other enteropathogens. When probed with sera from patients infected with *S. flexneri*, only 8, 2, and 3 antigenic bands reacted specifically against IgA, IgM, and IgG. The band at 21 kDa was recognized by IgA and IgG; that at 22 kDa was recognized only by IgG; those at 38, 48, 50, 52, and 60 kDa were recognized only by IgA; the 58 kDa band was recognized by IgA and IgM; and the 35 kDa band was recognized by IgA, IgM, and IgG. Table 1 shows a summary of the antigenic bands that reacted only with *Shigella flexneri* infection sera and not with other infection sera tested. Referring to the table, the 21 kDa protein band is suspected to be specific when detected with anti-IgA and anti-IgG; the 22 kDa band is suspected to be specific only when detected with anti-IgG; the 38, 48, 50, 52 and 60 kDa bands are suspected to be specific only for anti-IgA while the 58 kDa band is suspected to be specific only for anti-IgA and anti-IgM. Only the 35 kDa band was suspected to be specific when detected with all three classes of antibodies tested, *i.e.*, anti-IgA, anti-IgM and anti-IgG (Fig. S1).

This study was performed to determine the antigenic and specific proteins when reacted with *Shigella flexneri* infection serum and other infection serum. The results have been summarized as shown in Table 1. Referring Table 1, there were 31 surface protein bands of

**Table 1 Summary of the antigenic protein bands that reacted with sera from patients infected with *S. flexneri* and did or did not cross-react with sera from patients infected with enteropathogenic *Escherichia coli*, *Campylobacter jejuni*, *Vibrio cholerae*, *Salmonella* Typhi, *Giardia lamblia*, or *Entamoeba histolytica*.**

| Antigenic protein (kDa) | IgA | IgM | IgG |
|---|---|---|---|
| 21 | + | − | + |
| 22 | − | − | + |
| 35 | + | + | + |
| 38 | + | − | − |
| 48 | + | − | − |
| 50 | + | − | − |
| 52 | + | − | − |
| 58 | + | + | − |
| 60 | + | − | − |

**Note:**
+ indicates no cross-reactivity of the antigens with other infections.
− indicates cross-reactivity of the antigens with other infections.

**Table 2 Summary of the response of *Shigella flexneri* infected serum compared with normal human serum to the surface protein of *Shigella flexneri* strain SF480.**

| Antibodies | Antigenic pathway (kDa) | React with the primary antibody in the lane | | | | | | | | | | | |
|---|---|---|---|---|---|---|---|---|---|---|---|---|---|
| | | 1 | 2 | 3 | 4 | 5 | 6 | 7 | 8 | 9 | 10 | 11 | 12 |
| **IgA** | 21 | + | − | + | − | − | − | − | − | − | − | − | − |
| | 35 | + | + | − | − | − | − | − | − | − | − | − | − |
| | 38 | + | + | + | − | − | − | − | − | − | − | − | − |
| | 48 | + | − | − | − | − | − | + | − | − | − | − | − |
| | 50 | + | + | − | − | + | − | − | − | − | − | − | + |
| | 52 | + | + | + | − | + | − | − | − | − | − | − | + |
| | 58 | + | + | − | − | + | − | − | − | − | − | − | − |
| | 60 | + | + | + | − | − | − | − | − | − | − | − | − |
| IgM | 35 | + | + | − | − | − | − | − | − | + | − | − | − |
| | 58 | + | + | − | − | − | − | + | + | + | − | − | + |
| IgG | 21 | + | + | + | + | + | + | − | − | − | + | + | + |
| | 22 | + | + | + | + | + | + | + | + | + | + | + | + |
| | 35 | + | + | + | + | + | − | − | − | + | + | + | + |

**Note:**
+ indicates no cross-reactivity with the primary antibody in the lane.
− indicates cross-reactivity with the primary antibody in the lane.
Lanes 1 to 12 show the reaction of *Shigella flexneri* infected serum and normal human serum against the surface protein of *Shigella flexneri* strain SF480. Lane 1: *Shigella flexneri* SF480 infection serum, lane 2: *Shigella flexneri* infection serum (pooled), lane 3: normal human serum N1, lane 4: normal human serum N2, lane 5: normal human serum N3, lane 6: normal human serum N4, lane 7: normal human serum N5, lane 8: normal human serum N6, lane 9: normal human serum N7, lane 10: normal human serum N8, lane 11: normal human serum N9 and lane 12: normal human serum N10.

*Shigella flexneri* strain SF480 that were antigenic when reacted with *Shigella flexneri* infection serum. The antigenic protein has a molecular weight between 15 to 110 kDa.

Western blotting performed using human anti-IgA secondary antibody showed that there were eight surface protein bands of *Shigella flexneri* strain SF480 that only reacted

with *Shigella flexneri* infection serum (Table 2 lane 1 to lane 3) and did not cross-react with other infection sera (Table 2 lane 4 to lane 9). The protein has a molecular weight of 21, 35, 38, 48, 50, 52, 58 and 60 kDa (marked). Whereas, bands at molecular weight 15, 19, 22, 25, 27, 28, 30, 33, 36, 39, 42, 45, 56, 65, 76, 80, 86, 90, 95, 105 and 110 kDa were found to react cross-matched with several other infection sera tested. Therefore, the potential and suspected specific antigenic bands when detected with IgA antibodies are molecular weights of 21, 35, 38, 48, 50, 52, 58 and 60 kDa.

Western blot results using human anti-IgM secondary antibody showed that there were two bands of surface protein of *Shigella flexneri* strain SF480 which only reacted with *Shigella flexneri* infection serum (Table 2 lane 1 to lane 3) but not with other infection serum (Table 2 lane 4 to lane 9). The two protein bands have a molecular weight of 35 and 58 kDa. Whereas, molecular weight band 15, 19, 21, 22, 25, 27, 28, 30, 33, 36, 38, 39, 42, 45, 48, 50, 52, 56, 60, 65, 76, 80, 86, 90, 95, 105 and 110 kDa were found to cross-react with several other infection sera tested. Therefore, the potential and suspected specific antigenic bands for IgM antibodies are molecular weights of 35 and 58 kDa.

The results of the reaction with human anti-IgG secondary antibody showed that there were three bands of surface protein of *Shigella flexneri* strain SF480 which reacted with only *Shigella flexneri* infection serum (Table 2 lane 1 to lane 3) but not with other infection serum (Table 2 lane 4 to lane 9). The band has a molecular weight of 21, 22 and 35 kDa. Whereas, molecular weight band 15, 19, 25, 27, 28, 30, 33, 36, 38, 39, 42, 45, 48, 50, 52, 56, 58, 60, 65, 76, 80, 86, 90, 95, 105 and 110 kDa were found to cross-react with most other infection sera tested. Therefore, the potential and suspected specific antigenic bands for IgG antibodies are of molecular weight 21, 22 and 35 kDa.

## The response of *Shigella flexneri* infected serum compared with normal human serum to the surface protein of *Shigella flexneri* strain SF480

Based on the study made, some antigenic and suspected specific proteins have been identified. The protein has a molecular weight of 21, 22, 35, 38, 48, 50, 52, 58 and 60 kDa. The study in this section aims to further examine the specificity of proteins that have been identified in previous studies. The specificity of the protein was further determined by reacting *Shigella flexneri* infected serum with normal human serum. The results obtained are as shown in Fig. S2 and summarized as shown in Table 2.

The results in Fig. S2 and Table 2 in this study show that the bands suspected to be specific for IgA in the initial study, which are at molecular weights of 21, 38, 48, 50, 52, 58 and 60 kDa, were found to cross-react with normal human serum tested with anti-human IgA (lane 3 to lane 12). Based on Table 2, when tested with anti-IgA, the molecular weight bands of 21, 38 and 60 kDa cross-reacted with a normal human serum which is N1 serum, the 48 and 58 kDa bands also cross-reacted with a normal human serum which is each serum N5 and N3. The 50 kDa band cross-reacted with two normal human serums, namely N3 and N10, while the 52 kDa band cross-reacted with three normal human serums, namely N1, N3 and N10. Bands with molecular weights of 21 and 48 kDa only reacted with one serum of shigellosis patients, namely SF480 serum and did not react with
'pooled' shigellosis serum. The results in this study also show that the 35 kDa band reacts with both shigellosis patient serum tested and does not cross-react with normal human serum tested.

The results in this study also showed that proteins previously suspected to be specific for IgM antibodies at molecular weights of 35 and 58 kDa cross-reacted with normal human serum tested (Table 2). The 35 kDa band cross-reacted with one normal human serum tested, namely human normal serum N7 (lane 9) while the 58 kDa band cross-reacted with 4 of the normal human serum tested, namely human normal serum N5, N6, N7 and N10 (lane 7, 8, 9 and 12).

Protein bands suspected to be specific for IgG antibodies namely 21, 22 and 35 kDa proteins also cross-reacted with normal human serum IgG antibodies tested. The band at 21 kDa cross-reacted with seven normal human serums tested namely N1, N2, N3, N4, N8, N9 and N10 (lanes 3, 4, 5, 6, 10, 11 and 12). The 22 kDa band cross-reacted with all normal human serums tested while the 35 kDa band cross-reacted with seven normal human serums tested, namely N1, N2, N3, N7, N8, N9 and N10 (lanes 3, 4, 5, 8, 9, 11 and 12). Therefore, in the current study's further investigations on 35 KDa, it was found that the protein at 35 kDa was more specific when detected with human anti-IgA.

## The response of stool suspensions of *Shigella flexneri* infection compared with stool suspensions of other infections (for which the causative pathogen is known) to the surface protein of *Shigella flexneri* strain SF480

An antigenic and specific surface protein was detected when reacted with *Shigella flexneri* infection serum with human anti-IgA. The protein has a molecular weight of 35 kDa. The study in this section aims to determine whether antibodies against the 35 kDa protein are secreted into intestinal secretions and further to study the specificity of the 35 kDa protein when reacted with antibodies in the stool suspension of *Shigella flexneri* infection compared to the stool suspension of other pathogens. The presence of antibodies in the feces against *Shigella* infection can be detected by using the Western blot method. The first strip in each set in this study was incubated with the serum of *Shigella flexneri* infection as a control, the second strip was incubated with the stool suspension of *Shigella flexneri* infection and the next strip was incubated with the stool suspension of another infection known to be the causative pathogen. The results of this experiment are shown in Fig. S3 and have been summarized as in Table 3.

From the observations in the Fig. S3 and the Table 3, using human anti-IgA secondary antibodies, it was found that the 35 kDa protein can be clearly seen when reacted with *Shigella flexneri* infection serum (SF480) and *Shigella flexneri* infection stool suspension (SF10) but cannot be detected when reacted with faecal suspensions of other pathogens tested (Table 3). Meanwhile, the reaction with the IgM secondary antibody showed that the 35 kDa protein was only detected in the band that reacted with the *Shigella flexneri* infection serum but was not detected when reacted with the *Shigella flexneri* infection stool suspension and also the stool suspension of other pathogens tested. The results of the reaction with IgG secondary antibodies also gave the same results as IgM, that is, the 35

**Table 3 Summary of the reaction of the 35 kDa protein band with serum and stool suspensions of *Shigella flexneri* infections as well as stools of other infections (for which the causative pathogen is known).**

| Serial # | Antibody primer | Reaction of the 35 kDa protein with secondary antibodies | | |
|---|---|---|---|---|
| | | Anti-IgA | Anti-IgM | Anti-IgG |
| 1 | Serum shigelosis SF480 | + | + | + |
| 2 | Stool *Shigella flexneri* SF10 | + | – | – |
| 3 | Stool enteropatogenik *E. coli* 1 | – | – | – |
| 4 | Stool *Salmonella typhi* 1 | – | – | – |
| 5 | Stool *Salmonella typhi* 2 | – | – | – |
| 6 | Stool *Vibrio cholerae* | – | – | – |
| 7 | Stool *Campylobacter jejuni* | – | – | – |
| 8 | Stool enteropatogenik *E. coli* 2 | – | – | – |
| 9 | Stool *Entamoeba histolytica* | – | – | – |
| 10 | Stool *Giardia lamblia* | – | – | – |

**Note:**
+ indicates no cross-reactivity of the 35 kDa protein with secondary antibodies.
– indicates cross-reactivity of the 35 kDa protein with secondary antibodies.

kDa protein was only detected in the band that was reacted with *Shigella flexneri* infection serum and was not detected when reacted with *Shigella flexneri* infection stool suspension and also the stool suspension of other pathogens tested.

To test the immunoreactivity of these SAP antigens to intestinal secretions, the membrane strips were probed with freshly prepared stool suspensions from patients infected with *S. flexneri* and other common enteropathogens (Fig. 2 and Table 4). The majority of the SAP antigens cross-reacted with IgA in stool suspensions of patients infected with *Shigella* and other enteropathogens. Only the 35 kDa protein reacted specifically with IgA in all of the *S. flexneri* stool suspensions and did not cross-react with any of the stool samples of patients infected with other enteropathogens.

## DISCUSSION

The quickest approaches for precisely identifying bacterial infections are immunological procedures (*Jalal et al., 2022*; *Kortright et al., 2022*; *van der Put et al., 2022*). These tests use an antibody or antigen that exclusively identifies a certain bacterial species or set of species. Compared to conventional culture approaches, detection is now faster, more practical, sensitive, and precise thanks to recent advancements in immunoassay technology (*Mohammad Shabani et al., 2022*; *Shabani et al., 2022*). The design of several antibody tests has been made easier by the highly specific binding of antibody to antigen as well as the ease and adaptability of this response (*Kaminski et al., 2019*; *van der Put et al., 2022*). The majority of quick techniques used today for bacterial identification are antibody-based tests (*Boero et al., 2023*).

One of the most significant etiological agents of dysentery for residents of developing countries and tourists from industrialised countries is *Shigella* spp. (*Liu et al., 2021*; *Wang et al., 2022*). The infection often is fatal, particularly in young, malnourished children, if

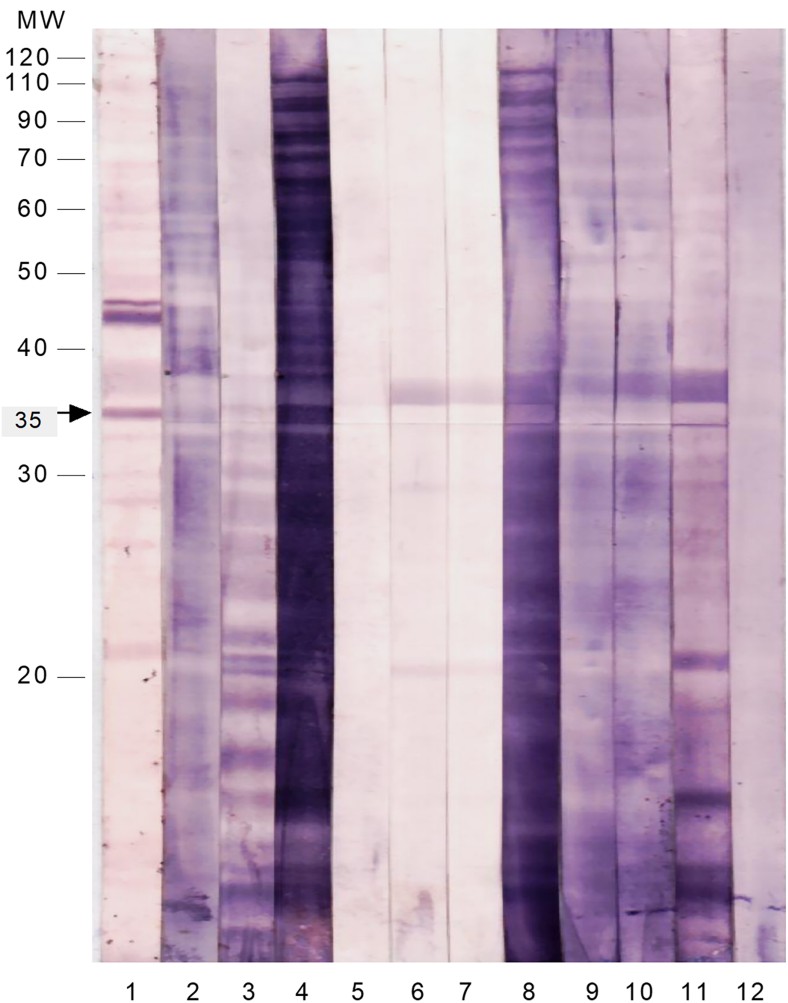

**Figure 2** **Immunoproteomics of the cell surface associated proteins of *S. flexneri* reacted with stool suspension from infected patients.** The membrane strips were probed with anti-human IgA and the following samples as 1st antibody: Lane 1: serum of *S. flexneri*; Lane 2: stool suspension of *S. flexneri* SF3; Lane 3: stool suspension of *S. flexneri* SF6; Lane 4: stool suspension of *S. flexneri* SF10; Lane 5: stool suspension of enteropathogenic *E. coli* 1; Lane 6: stool suspension of *Salmonella* Typhi 1; Lane 7: stool suspension of *Salmonella* Typhi 2; Lane 8: stool suspension of *Vibrio cholerae*; Lane 9: stool suspension of *Campylobacter jejuni*; Lane 10: stool suspension of enteropathogenic *E. coli* 2; Lane 11: stool suspension of *Entamoeba histolytica*; Lane 12: stool suspension of *Giardia lamblia*.

prompt and appropriate treatment is not given. Children under the age of 5 are the main targets of the disease; they represent 69% of all episodes and 61% of all deaths (*Bhaumik et al., 2023*; *Kotloff et al., 2018*). The purpose of this study was to identify specific and antigenic protein(s) that are common to *Shigella* spp. and that would be useful in the development of a rapid serodiagnostic test for shigellosis, particularly one that could be run using a stool sample rather than serum.

In our immunoproteomic analysis, probing with sera obtained from patients with shigellosis revealed that several SAP proteins were recognized by the host humoral response during infection. SAP proteins are the prime candidates for recognition by host

**Table 4 Summary of the immunoreactivity of the 35 kDa protein with serum and stool suspensions from patients infected with *S. flexneri* and with stool suspensions from patients infected with other enteropathogens.**

| Sample type | Immunoreactivity of the 35 kDa protein with anti-human IgA |
| --- | --- |
| Serum *S. flexneri* | + |
| Stool *S. flexneri* SF3 | + |
| Stool *S. flexneri* SF6 | + |
| Stool *S. flexneri* SF10 | + |
| Stool enteropathogenic *E. coli* 1 | − |
| Stool *Salmonella Typhi* 1 | − |
| Stool *Salmonella Typhi* 2 | − |
| Stool *Vibrio cholerae* | − |
| Stool *Campylobacter jejuni* | − |
| Stool enteropathogenic *E. coli* 2 | − |
| Stool *Entamoeba histolytica* | − |
| Stool *Giardia lamblia* | − |

**Note:**
+ indicates no immunoreactivity of the 35 kDa protein with anti-human IgA.
− indicates immunoreactivity of the 35 kDa protein with anti-human IgA.

antibodies because of their exposed location on the cell membrane (*Lavu, Mondal & Ramlal, 2021*; *Wu et al., 2019*). Previous studies of the host humoral response during *Shigella* infection investigated lipopolysaccharides, outer membrane proteins, and invasion plasmid-coded proteins (*Brengi et al., 2019*; *Taneja et al., 2011*; *Zhang et al., 2021*). The current study was the first to report immunoproteomic data for SAP proteins in *Shigella*.

In the current study, most of the sera from healthy individuals cross-reacted with the SAP proteins of *Shigella*, which suggests or assumes that members of the population in the region tested have a history of shigellosis. A specific 35 kDa antigenic protein was detected in all four species of *Shigella*, and it was specific to *Shigella* and did not show cross-reaction with sera from infections with other enteropathogens. Therefore, the existence of antibodies specific to it in sera from patients with dysentery means that they are infected with one of the *Shigella* species. The 35 kDa protein would thus be a specific diagnostic test for shigellosis, indicating that the antibodies found in the shigellosis sera had been acquired as a consequence of a particular infection.

To determine if the presence of this 35 kDa immunogenic band was also specific to the mucosal secretions of shigellosis patients, we tested stool samples obtained from patients infected with various other enteropathogens (Table 2). None of these stool suspensions cross-reacted with the 35 kDa protein when probed against IgA, which suggests that it is indeed secreted into the mucosal secretion of patients with shigellosis. Detection of antibodies against *Shigella* in mucosal secretions in dysenteric patients was previously shown only for invasion plasmid antigen and lipopolysaccharide antigens (*Sheikh et al., 2022*; *Xerri & Payne, 2022*).

Our experiments showed that the 35 kDa protein from *S. sonnei*, *S. dysenteriae*, *S. flexneri*, and *S. boydii* reacted specifically against IgA present in both sera and stool suspensions from patients. Because this protein is present in all *Shigella* spp., immunodetection of this protein directly in stool specimens could be a simple approach to identifying all species of this pathogen. A similar strategy used in a previous study identified a specific antigenic protein for *Salmonella* Typhi (*Islam, Singh & Ismail, 2011*). The findings from that study led to the development of *Typhidot* and *Typhidot-M*, which currently are being used as routine diagnostic tests for this species in many laboratories worldwide (*Safi et al., 2023*).

**Study limitations:** Despite the current study provides a through data on 35 KDa isolated from *S. flexneri*, the current study has certain limitations. At first, in the current study we could not estimate the level and retention time of detectable IgA antibodies against the 35 kDa protein in the stools of infected patients, which could be vital to defend the study outcomes. Secondly, the current study could not be studied on other enteropathogens which can provide more good explanations on SAP.

## CONCLUSIONS

In comparison to the present standard technique, the use of a particular antigen in the diagnosis of shigellosis offers numerous benefits. The creation of a quick, accurate, and sensitive diagnostic test for shigellosis would aid in early clinical assessment and provide a means of measuring the disease's epidemiological effects. The current study provides a detailed work with potentials to conduct further large-scale studies.

## ACKNOWLEDGEMENTS

The authors would like to thank the Hospital Universiti Sains Malaysia, Malaysia for providing the isolates and clinical samples.

### Funding

This work was supported by the Universiti Sains Malaysia Research University Grant (Molecular approaches to fundamental studies on specific biomarkers of Shigella and development of sustainable rapid nano-biodiagnostics for shigellosis; Grant number 1001/PSKBP/8630012). The funders had no role in study design, data collection and analysis, decision to publish, or preparation of the manuscript.

### Grant Disclosures

The following grant information was disclosed by the authors:
Universiti Sains Malaysia Research University: 1001/PSKBP/8630012.

### Competing Interests

The authors declare that they have no competing interests.

## Author Contributions

- Kirnpal Kaur Banga Singh conceived and designed the experiments, performed the experiments, prepared figures and/or tables, authored or reviewed drafts of the article, and approved the final draft.
- Mohd Zaki Salleh conceived and designed the experiments, performed the experiments, authored or reviewed drafts of the article, and approved the final draft.
- Naveed Ahmed analyzed the data, prepared figures and/or tables, authored or reviewed drafts of the article, and approved the final draft.
- Chan Yean Yean analyzed the data, authored or reviewed drafts of the article, and approved the final draft.
- Asma Ismail conceived and designed the experiments, authored or reviewed drafts of the article, and approved the final draft.

## Human Ethics

The following information was supplied relating to ethical approvals (*i.e.*, approving body and any reference numbers):

The study was conducted in accordance with the Declaration of Helsinki and approved by the Human Research Ethics Committee of Universiti Sains Malaysia (protocol code: US-MKK/PPSP/JEPeM [248.3(10)]) for studies involving human samples.

## Data Availability

The raw data are available in the Supplemental Files.

## Supplemental Information

Supplemental information for this article can be found online at http://dx.doi.org/10.7717/peerj.17498#supplemental-information.

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
