# Peer review of "Identification and analysis of immunoreactive proteins of Shigella flexneri in human sera and stool specimens"

_PeerJ, doi:10.7717/peerj.17498_

## Round 0.1 · original submission · Major Revisions

Please address the concerns of all reviewers and amend the manuscript accordingly.

**Language Note:** The review process has identified that the English language must be improved. PeerJ can provide language editing services - please contact us at [email protected] for pricing (be sure to provide your manuscript number and title). Alternatively, you should make your own arrangements to improve the language quality and provide details in your response letter. – PeerJ Staff

·

Basic reporting

In this study, the authors have analyzed the immunogenic potency of Surface Antigenic Proteins (SAP) from Shigella flexneri and discovered a 35 kDa SAP protein that could potentially be used for a rapid and efficient diagnostic assay for shigellosis. The efforts of the authors are appreciated and the following are my comments on the minor side:

1. The authors may clarify what the term "surface protein extract time" refers to. The time periods mentioned in the results section (Line: 149) and in Figure 1 refer to which step in the SAP extraction
Apart from these, listed below are my minor concerns that may be appropriately addressed:

1. Line 37 and 38: The line that speaks about the immunoreactivity of the 35kDa SAP from S. flexneri against other Shigella spp. is incomplete.

2. Line 38-39: "This is the first report demonstrating the presence of immunoreactive Shigella SAPs in stool suspensions." Instead, does the author mean the detection of IgA antibodies specific to the 35kDa SAP from S. flexneri from the stool?

3. Line 47: Repetition of Shigella is found. The line could be made as "S. sonnei, S. flexneri, S. boydii, and S. dysenteriae."

4. Line 48: Unlike mentioned by the authors, 90% of global shigellosis is attributed to S. sonnei and S. flexneri. Though S. dysenteriae 1 causes the most severe infection that leads to high fatality (20%), its incidence rate presently is very low. The references quoted in this section are found inappropriate.

5. Line 51 could be made a little clearer. I suppose the authors want to convey, "The molecular origins of Shigella pathogenesis have been extensively discussed previously on a large scale."

6. Line 53: "Shigella can get beyond these obstacles because to... antimicrobial peptides" remains incomplete.

7. Lines 65-67: Authors may redefine the lines from 65 to 67 that mention the need for blood to isolate Shigella. Since only during Shigella bacteremia will we need to isolate Shigella from blood.
I request the authors to rewrite the introduction section, quoting the appropriate references. "Correct punctuation and grammar, please."

Experimental design

The authors may clarify what the term "surface protein extract time" refers to. The time periods mentioned in the results section (Line: 149) and in Figure 1 refer to which step in the SAP extraction protocol.

Validity of the findings

1. Information on whether the 35kDa SAP isolated from Shigella flexneri strain SF480 could diagnose shigellosis attributed by S. sonnei, S. dysenteriae, and S. boydii as well. Similarly, the authors may provide information on whether the 35kDa protein is isolated from other Shigella spp. like S. sonnei, S. dysenteriae, and S. boydii could potentially respond to sera from Shigella flexneri infected patients.


2. Did the authors estimate the level and retention time of detectable IgA antibodies against the 35kDa protein in the stools of infected patients? This is vital since the study outcomes aim to escalate for diagnostic applications.

·

Basic reporting

1. More clarification is required for figures 1 and 2.
2. Line 239: Where are sections 3.2 and 3.3?
3. Table 2: Please correct 'Lanes 1 to 12'.

Experimental design

1. Line 233: Detailed writing is required about the further studies mentioned.
2. Line 111: Please write in detail about harvesting bacterial cells.

Validity of the findings

No comment

Additional comments

Some words should be italic.

Reviewer 3 ·

Basic reporting

Language needs to be revised throughout, especially in the introduction where some grammatical errors were noticed.

Shigella should be italicised throughout the article. Please revise.

Line 53 - please revise sentence.

Please conform to journal standards. Subheadings are to be at the beginning of the paragraph. This was not followed in the methods section.

Field background/context were provided, but misleading statements were noted in some areas.

Line 50 - Please add references that review the origin of Shigella pathogenesis.

Line 65 - How prevalent is Shigellosis in children? The authors' statement is not true globally. Although it may be the most common cause of diarrheal illness in some locations (e.g. Africa, South Asia), it is not necessarily highly prevalent. Please provide a reference and/or supporting information for this statement, or remove as it is misleading.

Line 179, Line 187, Line 195 - please add which figures you are referring to in the text. Lanes are mentioned, but it is unclear which figure is being described.

Line 198 - is Table 3.4 referring to Table 3? If so, Table 3.4 is mentioned before Table 2 in the text. Please list tables in order of appearance in text.

There is no description of the +/- annotations in the footnotes of Table 2 nor Table 3. Please add for clarification.

Experimental design

Article is within aims and scope.

There is no approval reference number from the Institutional Review Board. Line 77 lists a protocol code, is this the approval reference?No written approval supplemental files were attached

The authors state that written consent has been obtained from the patients. An empty copy of the consent form/information sheet has not been provided as a supplemental file as is required by the journal.

Investigation would benefit from a larger panel to determine inclusivity/exclusivity of the method. Only 8 other (non-shigella) enteropathogens were tested in this study, which is not inclusive.

Did the authors also test human serum from infections with other pathogens? Or just normal human serum and shigella-infected serum?

Validity of the findings

As with the introduction, I believe some findings in the discussion are misleading or incorrect.

Figure 2 - although the 35 kDa protein is clearly visible in lane 1, the serum of S. flexneri, I do not clearly see this band in any of the other lanes including the stool suspensions from S. flexneri infection. Lane 4 does have some darkening in this area, but overall the banding is not clearly distinct and may not be easily readable/detected by someone using this test to diagnose or positively identify presence of S. flexneri. I believe the study would benefit from investigating additional processing of stool samples, or the authors should discuss future plans/steps to address this.

Line 278 - sentence is false and misleading. The reference cited here for Yang compared molecular (PCR) techniques and makes no mention of antibody-based tests. Although the reference cited here for Necchi did use immune-detection based methodology, it was for determining antigen content in vaccine products and not used as a diagnostic test for stool or serum samples, therefore should not be cited as being a quick technique for bacterial identification. Remove or revise this section.

Line 296 - How can you be sure that most of the individuals have a history of shigella if you have not exhausted other possibilities? How do you know it is not non-specific cross reactivity with something else found in stool, or another pathogen? I don’t believe you have shown this to be true in this manuscript. Statement should be removed.

Line 298 - The authors did not show cross-reaction with sera from other enteropathogens test in this study only. Not all enteropathogens were investigated. A larger inclusivity/exclusivity analysis is required in order to make this statement that the findings are in fact "specific". Please revise wording.

Line 300 - same as above. Would benefit to include additional genera, species, and serovars/serotypes in the exclusivity testing. Therefore, you can not yet state this work as a specific diagnostic test, only that it has the potential following further study.

Additional comments

Article is interesting. Data is provided. Please revise to ensure that "blanket" statements regarding specificity of this technique, as this study is not inclusive/exclusive enough to define it as a specific method.
Also, please revise text for misleading statements regarding prevalence of Shigellosis, and review references to ensure they are backing up statements.

---

## Round 0.2 · accepted · Accept

In my view, all the issues pointed out by the reviewers were addressed and the manuscript was revised accordingly. The amended version is acceptable now.